# Physiological Effects of Low Salinity Exposure on Bottlenose Dolphins (*Tursiops truncatus*)

**Abby M. McClain** [1,*], **Risa Daniels** [1], **Forrest M. Gomez** [1], **Sam H. Ridgway** [1], **Ryan Takeshita** [1], **Eric D. Jensen** [2] and **Cynthia R. Smith** [1]

1   National Marine Mammal Foundation, 2240 Shelter Island Dr. Suite 200, San Diego, CA 92106, USA; risa.daniels@nmmf.org (R.D.); forrest.gomez@nmmf.org (F.M.G.); sam.ridgway@nmmf.org (S.H.R.); ryan.takeshita@nmmpfoundation.org (R.T.); cynthia.smith@nmmf.org (C.R.S.)
2   U.S. Navy Marine Mammal Program, 4301 Pacific Highway, San Diego, CA 92110, USA; eric.d.jensen@navy.mil
*   Correspondence: abby.mcclain@nmmf.org

**Abstract:** Bottlenose dolphins (*Tursiops truncatus*) have a worldwide distribution in temperate and tropical waters and often inhabit estuarine environments, indicating their ability to maintain homeostasis in low salinity for limited periods of time. Epidermal and biochemical changes associated with low salinity exposure have been documented in stranded bottlenose dolphins; however, these animals are often found severely debilitated or deceased and in poor condition. Dolphins in the U.S. Navy Marine Mammal Program travel globally, navigating varied environments comparable to those in which free-ranging dolphins are observed. A retrospective analysis was performed of medical records from 46 Navy dolphins and blood samples from 43 Navy dolphins exposed to a variety of salinity levels for different durations over 43 years (from 1967–2010). Blood values from samples collected during low salinity environmental exposure (salinity ranging from 0–30 parts per thousand (ppt) were compared to samples collected while those same animals were in a seawater environment (31–35 ppt). Epidermal changes associated with low salinity exposure were also assessed. Significant decreases in serum sodium, chloride, and calculated serum osmolality and significant increases in blood urea nitrogen and aldosterone were observed in blood samples collected during low salinity exposure. Epidermal changes were observed in 35% of the animals that spent time in low salinity waters. The prevalence of epidermal changes was inversely proportional to the level of salinity to which the animals were exposed. Future work is necessary to fully comprehend the impacts of low salinity exposure in bottlenose dolphins, but the physiological changes observed in this study will help improve our understanding of the upper limit of duration and the lower limit of salinity in which a bottlenose dolphin can maintain homeostasis.

**Keywords:** salinity; bottlenose; dolphin; physiology; freshwater; osmoregulation; skin

## 1. Introduction

Bottlenose dolphins (*Tursiops truncatus*) are observed worldwide in temperate and tropical waters, and certain populations are known to inhabit bays or estuaries of brackish waters with salinities ranging from 15–25 ppt [1–3]. It has been well-documented that coastal dolphins may seek out freshwater or low salinity environments to feed and forage; for example, dolphins in northern Florida often swim into the St. Johns River to feed for periods of a week or more, exposing themselves to salinities even lower than estuaries (as low as 0 ppt) [1,4]. Dolphins inhabiting Barataria Bay in the northern Gulf of Mexico have been found to inhabit environments with salinities ranging between 1.6–32.08 ppt, spending between 1–12 consecutive days at salinities below 7.89 ppt [2,5]. Thus, it is likely these

animals have homeostatic mechanisms that allow them to be in freshwater for periods of time despite their evolutionary adaptation toward conservation of freshwater [6–8].

Marine vertebrates have evolutionary physiologic adaptations to counter-act a hyper-osmotic environment and avoid dehydration [9]. Some species of marine teleosts actively drink seawater to replace water that is lost osmotically. In these species, the majority of water is absorbed via the gastrointestinal epithelium with the excess salt excreted and regulated by the renin–angiotensin system (RAS) [9]. Similar to teleost fish, elasmobranchs use the RAS to regulate their electrolyte physiology; however, elasmobranchs use organic osmoles like urea to maintain their plasma osmolality to slightly above that of seawater [9]. Other aquatic species, like West Indian manatees (*Trichechus manatus*), do not voluntarily drink seawater. Manatees, which inhabit both freshwater and marine habitats, will drink fresh water when it is available to maintain water balance [9,10]. Studies have shown that high aldosterone levels persist in wild freshwater manatees, indicating a need to conserve sodium [10]. Knowledge pertaining to the drinking behaviors of cetaceans is sparse, but studies using isotopic dilution techniques have shown that cetaceans do consume a small amount of seawater each day, either passively through foraging, or actively through drinking [8,9,11]. The RAS system and aldosterone are present and active in dolphins and other marine mammals, similar to terrestrial animals. The reniculate kidney of the cetacean excretes more water and salts than is typical of most terrestrial mammals [9,10,12,13].

Dolphins have the physiologic capabilities to regulate their plasma electrolytes and osmolality when consuming seawater [14]. In an earlier freshwater exposure study, Ridgway [7,8] found that dolphin's urine electrolytes rapidly declined within hours of freshwater exposure, while serum electrolytes remained within the normal reference range. Upon return to a seawater environment (34 ppt), urine electrolytes and osmolality increased up to 10-fold. The stability of blood electrolytes and the adaptation of urine electrolyte output observed in this study showed that dolphins could maintain homeostasis in freshwater. Freshwater intoxication in other species, including humans, results in hyponatremia [15]. Hyponatremia is diagnosed as a serum sodium <135 mEq/L in humans and as either acute or chronic. Acute hyponatremia occurs when the serum sodium level decreases below 135 mEq/L in less than 48 h; chronic hyponatremia is a decrease in serum sodium that develops over a time period greater than 48 h [15]. Acute hyponatremia presents with clinical neurological signs including seizures, impaired mental status, coma, or death [16,17]. Chronic hyponatremia often lacks clinical signs as the body adapts by producing particles from cellular metabolism called idiogenic osmoles to counteract a decreased sodium level [16].

Some knowledge pertaining to physiological effects associated with dolphins' exposure to low salinity waters has been obtained from stranding data, where the duration and level of salinity exposure is often unknown or estimated [6]. Commonly, the diagnosis of freshwater intoxication relies upon the visualization of skin lesions in combination with the salinity level measured in the stranded dolphin's location [6,18–21]. The skin lesions observed in dolphins secondary to low salinity exposure are often due to hydropic degeneration (ballooning degeneration) of the superficial epithelial cells [3,18,22]. A recently published study described skin lesions from a single dolphin exposed to freshwater for 32 days as diffusely pale grey and sloughing, with superficial erosive lesions on the flukes, head, mid-epaxial region, and dorsal fin [3]. Histologically, the lesions showed "epithelial erosion, edema, hydropic degeneration of the superficial epithelial cells, and separation of superficial epithelial layers with serum accumulation" [3]. Skin lesions are significant to dolphin health as a net gain of freshwater can occur through compromised skin when dolphins are exposed to hypo-osmotic environments [9]. Changes in blood variables have also been documented from stranded animals exposed to low salinity waters [3,6]. Significant differences in serum electrolytes (sodium and chloride) and serum bicarbonate were observed in animals stranded in low salinity water; however, the duration in which the animal was exposed to the low salinity environment was often estimated based off of previous focal sightings [6]. Mild hyponatremia and hypochloremia were also found in a single dolphin exposed to freshwater for 32 days [3]. Unusual mortality events in bottlenose dolphins have

been associated with record rainfalls and an increase in freshwater runoff in which increased mortality was likely associated with a combination of low salinity exposure and chemical contamination from the runoff [23]. In 2017 a record-breaking rainfall occurred during hurricane Harvey that plummeted the salinity in the Galveston Bay estuary from 14 ppt to less than 1ppt resulting in an increase in low salinity skin lesion prevalence in the resident dolphins from 49% to 96% [24]. Morbidity and mortality associated with low salinity exposure in bottlenose dolphins is likely a combination of biochemical disturbances and compromised epidermal integrity. Currently, the lower limit of salinity and the upper limit of exposure time for bottlenose dolphins in low salinity environments are unknown.

With growing concern for constantly changing ecosystems associated with climate change, increased rainfalls, and human-associated threats that have the potential to increase freshwater influx from rivers, there is an urgency to better understand the physiologic impacts of low salinity exposure on bottlenose dolphins [25,26]. Through the U.S. Navy Marine Mammal Program (MMP), bottlenose dolphins are deployed globally and are monitored closely as part of a comprehensive health program. Since these animals may live in a variety of environments with varied salinity, routine health monitoring provides unique data for the better understanding of the effects of varied salinity on dolphins. The aim of this retrospective study was to establish whether exposure to a variety of natural salinity environments, ranging from fresh water to seawater (31–35 ppt), for varying ranges of time had physiological impacts on bottlenose dolphins. For the purpose of this study, we compared biomarkers commonly associated with freshwater exposure from dolphins in a seawater environment (>30 ppt) to dolphins in low salinity environments (0–10 ppt, 11–20 ppt, and 21–30 ppt) to determine the effects of varying salinity levels and duration of exposure on bottlenose dolphins [3,6,10].

## 2. Materials and Methods

### 2.1. Study Design

A retrospective review of data from Navy dolphins that inhabited environments with salinities below 30 ppt for varying amounts of time was performed and compared to control data collected from the same dolphins inhabiting water with salinities >30 ppt. Samples from Navy dolphins were collected during their routine comprehensive care and under U.S. Code, Title 10, USC 7524 authority. Samples were opportunistic and collected as part of the Navy's in-depth preventative medicine program; thus, an IACUC approval was not needed for this study. Approval by MMP for review and publishing of the data occurred on 2 February 2019. The dolphins involved in this retrospective analysis were deployed to different locations throughout the world. The health of the dolphins and their environment were closely monitored on a daily basis by animal care specialists and medical staff. If any adverse medical conditions occurred during these deployments, either secondary to the environment or otherwise, the dolphins were treated immediately, and environmental mitigation measures were utilized.

Historical records of 46 individual dolphins (female = 19, male = 27) exposed to varying salinity ranges at 10 different geographical locations between June 1967 and October 2010 were reviewed retrospectively. Records included written medical observations of skin condition from animal care and medical professionals, as well as blood biochemistry results. The dolphins were housed in above-ground deployable pools or free-floating ocean enclosures. The dolphins housed in free-floating ocean enclosures were typically located along the coastline or within harbors at each location. Salinity was measured at a minimum on a daily basis, but often multiple times per day at each location using a refractometer (Science First, Yulee, FL) as part of routine environmental monitoring. In order to stratify the data, salinity levels were separated into three categories: category 1, category 2, and category 3 (salinity between 0–10 ppt, 11–20 ppt, and 21–30 ppt, respectively). Each dolphin was assigned to one category for blood sample analysis based on the highest exposure level of salinity. For example, if a dolphin was exposed to levels of salinity ranging from 10–25 ppt, the animal was placed in category 3.

Blood samples were collected as part of the routine veterinary care and monitoring of the dolphins. Serum aldosterone and cortisol were also measured from six individual dolphins. Visual epidermal assessment was performed on all dolphins prior to, during, and after exposure to low salinity. Inclusion criteria for assignment of an individual dolphin to a specific salinity category required the dolphin to have only been exposed to salinity levels within that specific category (0–10 ppt, 11–20 ppt, and 21–30 ppt). Dolphins exposed to a variety of salinity levels across multiple categories were excluded from this analysis. Epidermal analysis was performed on this cohort of dolphins individually. A low salinity skin lesion was defined as an overall lightening or darkening of the skin color, heavy/abnormal sloughing of the epithelial layer of the skin with no evidence of ulcers or erosions, and/or erosive/ulcerative lesions on the skin with or without secondary infection [21,24,27]. An overall lightening or darkening of the skin was considered mild, heavy/abnormal sloughing was considered moderate, and ulcerative/erosive lesions were considered severe.

## 2.2. Sample Collection and Processing

Blood samples were collected under voluntary presentation or manual restraint from the periarterial venous rete (PAVR) on the ventral aspect of either fluke blade, or from the PAVR in the caudal peduncle with needle sizes ranging from 19–22 gauges. Blood was collected either via syringe or butterfly needle attached to a vacutainer collection system and placed into BD vacutainers (BD, Franklin Lakes, NJ). BD vacutainers containing serum separator were used for biochemical analysis. Blood collection in the 1960s and 1970s required blood to be placed in glass test tubes that were treated with silicone solution for biochemical analysis [28]. Due to the retrospective nature of the analysis and the large timeframe the study period covers, the collected biochemical blood samples were analyzed at multiple different laboratories throughout the world using automated chemistry analyzers. Aldosterone and cortisol samples were analyzed at Loma Linda University Medical Center (Loma Linda, CA) via radioimmunoassay (RIA).

The analyzed variables included serum sodium (mEq/L), chloride (mEq/L), potassium (mEq/L), phosphorous (mg/dL), blood urea nitrogen (BUN; mg/dL), creatinine (mg/dL), glucose (mg/dL), aldosterone (pg/mL), cortisol (mcg/dL), and calculated serum osmolality (cOsm/Kg). The calculated serum osmolality was performed utilizing the reported blood values according to the Worthley equation [29]:

$$\text{Osmolality} = (2[\text{Na}^+ \, (\text{mEq/L})]) + ([\text{Glucose (mg/dL)}]/18) + ([\text{BUN (mg/dL)}]/2.8). \tag{1}$$

## 2.3. Statistical Analysis

Data were analyzed using R v 3.6.1 [30] via RStudio v 1.2.5019 [31]. Statistical analyses were performed on blood variables at the population level. The effect of the salinity level to which the dolphins were exposed, and the duration of exposure were assessed. All data were assessed for normality prior to analysis; serum aldosterone was the only variable that did not have a normal distribution. Blood values obtained while dolphins were in lower salinity environments were compared to values obtained while dolphins were in marine environments (controls; 31–35 ppt) through mixed effect models and likelihood ratio tests for normally distributed variables using the lme4 [32] and lmerTest [33] packages in R, taking into account individual dolphin variation and location as random effects. Age was not significant for any of the biomarkers assessed except for serum potassium and chloride, and sex was not significant for any of the biomarkers assessed except for serum chloride. In those cases, age and/or sex were included as covariates in the final model for those biomarkers. A Wilcoxon rank sum test was used to analyze the effect of salinity on serum aldosterone and a generalized linear model with a gamma distribution was used to analyze the effect the duration of exposure to low salinity had on aldosterone. A Student's t-test was used to analyze the effect of salinity on serum cortisol.

## 3. Results

### 3.1. Blood Analysis Results

A total of 312 blood samples were obtained from 43 different dolphins (minimum of one sample, maximum of 45 samples from an individual dolphin) over the course of 10 different low salinity exposure events. The duration of low salinity exposure ranged from 3 to 79 consecutive days. Blood samples were collected in all seasons in both cold and warm temperature waters. Our analyses suggested that lower salinity exposure had the most influence on serum sodium, chloride, BUN, aldosterone, and calculated serum osmolality values, but had little effect on serum potassium, cortisol, glucose, phosphorus, or creatinine (Table 1). We do not suspect that water temperature had an effect on serum sodium, chloride, BUN, and calculated serum osmolality values; however, water temperature may have affected the changes in serum aldosterone that were observed.

**Table 1.** Likelihood ratio test results for each analyte and their associated p-values. Likelihood ratio tests were performed to assess the effects of the level of salinity exposure compared to controls on each analyte as well as the effects of the duration of exposure to a low salinity environment on each analyte. Individual animals and the location of each animal were considered random effects in the models. The asterisks (*) denote analytes that were significantly different from dolphins in low salinity environments compared to control dolphins.

| Analyte | Salinity | Duration |
|---|---|---|
| | *p*-Value | *p*-Value |
| Sodium | 0.001 * | 0.719 |
| Chloride | <0.001 * | 0.501 |
| Blood Urea Nitrogen | <0.001 * | 0.683 |
| Aldosterone | 0.018 * | 0.891 |
| Calculated osmolality | 0.043 * | 0.941 |
| Potassium | 0.130 | 0.570 |
| Phosphorus | 0.175 | 0.139 |
| Glucose | 0.927 | 0.109 |
| Creatinine | 0.081 | 0.901 |
| Cortisol | 0.263 | 0.075 |

Lower salinity had a strong effect on serum sodium and chloride levels, resulting in decreased values compared to dolphins in waters >30 ppt (ChiSq = 15.91, df = 3, $p = 0.001$; ChiSq = 46.5, df = 3, $p < 0.001$, respectively). The level of salinity was found to affect the magnitude of the electrolyte changes (Table 2). Serum sodium measured from dolphins exposed to a salinity level between 21–30 ppt was very similar to the serum sodium measured from dolphins exposed to a seawater environment. The larger differences in serum sodium occurred when dolphins were exposed to a salinity level of less than 20 ppt (Table 2). Interestingly, the largest serum chloride difference was found in dolphins exposed to a salinity of <10 ppt whereas the serum chloride in dolphins exposed to salinities of 11–20 ppt and 21–30 ppt were very similar to the serum chloride in control dolphins (Table 2). The duration of time animals were exposed to a low salinity environment did not have an effect on serum sodium (ChiSq = 0.149, df = 1, $p = 0.700$) or chloride (ChiSq = 0.292, df = 1, $p = 0.589$).

**Table 2.** The average change in the analyte when dolphins were exposed to a low salinity environment compared to the controls are depicted for each category of salinity (Category 1 = 0–10 ppt, Category 2 = 11–20 ppt, Category 3 = 21–30 ppt). Aldosterone was analyzed using a Wilcoxon rank sum test for a non-parametric distribution, and cortisol was analyzed using a Student's t-test for parametric distributions. The asterisks (*) denote analytes that were significantly different from dolphins in low salinity environments compared to control dolphins.

| Analyte | Units | Category | Salinity Range | Average Change in Analyte Compared to Controls | *p*-Value |
|---------|-------|----------|----------------|-----------------------------------------------|-----------|
| Sodium | mEq/L | 1 | 0–10 ppt | −3.85 | <0.001 * |
| | | 2 | 11–20 ppt | −3.34 | 0.018 * |
| | | 3 | 21–30 ppt | −0.79 | 0.674 |
| Chloride | mEq/L | 1 | 0–10 ppt | −8.43 | <0.001 * |
| | | 2 | 11–20 ppt | −0.33 | 0.830 |
| | | 3 | 21–30 ppt | −1.54 | 0.495 |
| Potassium | mEq/L | 1 | 0–10 ppt | −0.04 | 0.484 |
| | | 2 | 11–20 ppt | 0.09 | 0.388 |
| | | 3 | 21–30 ppt | 0.27 | 0.041* |
| Blood Urea Nitrogen | mg/dL | 1 | 0–10 ppt | 8.69 | <0.001 * |
| | | 2 | 11–20 ppt | 7.31 | 0.007 * |
| | | 3 | 21–30 ppt | 2.33 | 0.591 |
| Calculated osmolality | cOsm/Kg | 1 | 0–10 ppt | −7.26 | 0.012 * |
| | | 2 | 11–20 ppt | −4.33 | 0.142 |
| | | 3 | 21–30 ppt | −2.64 | 0.537 |
| Glucose | mg/dL | 1 | 0–10 ppt | 2.47 | 0.599 |
| | | 2 | 11–20 ppt | −0.07 | 0.735 |
| | | 3 | 21–30 ppt | 1.91 | 0.900 |
| Phosphorus | mg/dL | 1 | 0–10 ppt | 0.27 | 0.145 |
| | | 2 | 11–20 ppt | −0.58 | 0.106 |
| | | 3 | 21-30 ppt | −0.08 | 0.661 |
| Creatinine | mg/dL | 1 | 0–10 ppt | −0.02 | 0.959 |
| | | 2 | 11–20 ppt | −0.08 | 0.047 * |
| | | 3 | 21–30 ppt | −0.16 | 0.010 * |
| Aldosterone | pg/mL | 1 | 0–10 ppt | 3.1 times greater than seawater values | 0.018 * |
| Cortisol | mcg/dL | 1 | 0–10 ppt | 1.27 times greater than seawater values | 0.263 |

A lower salinity environment also had a strong effect on serum BUN levels, where dolphins exposed to lower salinity environments had elevated levels of BUN compared to controls (ChiSq = 19.92, df = 3, *p* < 0.001), most notably at salinities <20 ppt (Table 2). The duration of exposure to a low salinity environment did not affect serum BUN (ChiSq = 0.088, df = 1, *p* < 0.767).

Environment did not affect serum BUN Lower salinity waters influenced osmolality: calculated serum osmolality was lower in dolphins exposed to a low salinity environment compared to controls (ChiSq = 8.52, df = 3, *p* = 0.043), especially in waters <10 ppt (Table 2). The duration of low salinity exposure did not have an effect on serum osmolality (ChiSq = 0.419, df = 1, *p* = 0.518).

Serum aldosterone was elevated in dolphins exposed to a low salinity environment compared to controls (T = 51, *p* = 0.018). Serum aldosterone samples were obtained only from dolphins exposed to a salinity of <10 ppt (n = 16) and dolphins exposed to seawater (n = 13). The average serum aldosterone while animals were exposed to waters below 10 ppt was 24.24 pg/mL; the average serum aldosterone while animals were exposed to a seawater environment was 7.73 pg/mL. The duration of low salinity exposure did not have an effect on serum aldosterone (df = 1, *p* = 0.891).

During one specific event, pooled serum aldosterone samples were collected prior to (pre-exposure) and three days after (post-exposure) dolphins (n = 4) spent time in waters <10ppt. Aldosterone levels

were greater post-exposure (median = 12.72 pg/mL; mean = 20.05 pg/mL) than pre-exposure (median = 0.26 pg/mL; mean = 1.2 pg/mL) (Figure 1).

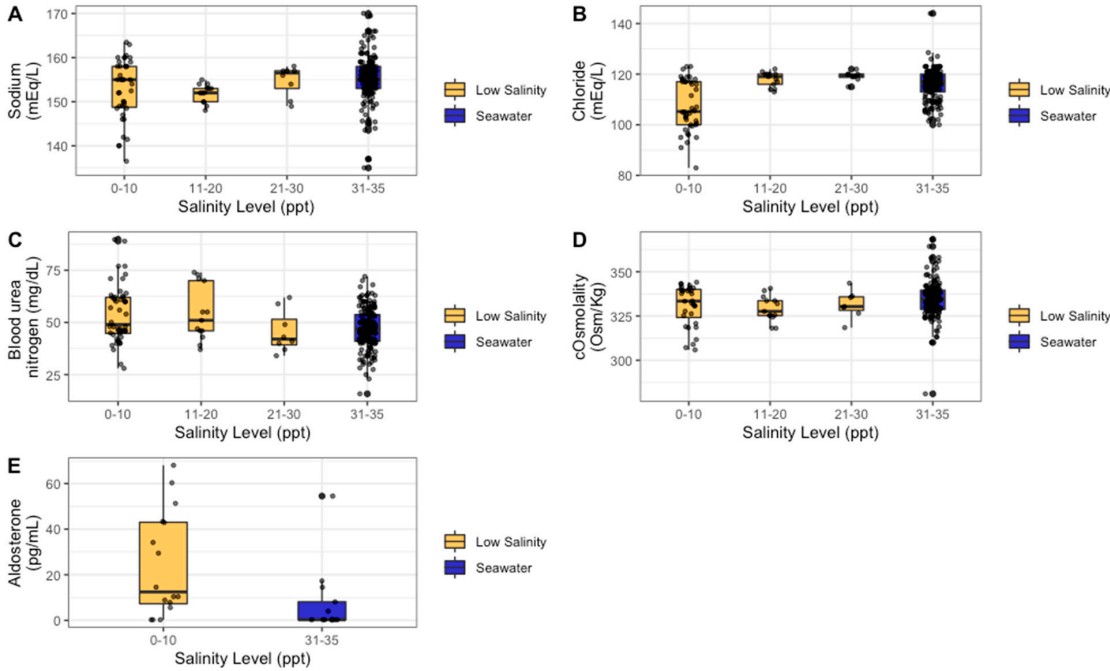

**Figure 1.** Boxplots of the significant variables in low salinity environments compared to controls. Actual values overlay the boxplots in black. (**A**) Serum sodium; (**B**) serum chloride; (**C**) serum blood urea nitrogen; (**D**) calculated serum osmolality; (**E**) serum aldosterone.

### 3.2. Epidermal Assessment

Skin assessments were performed prior to, during, and after exposure to low salinity in all dolphins. Lesions developed in 35% (n = 16) of dolphins and were first observed between 13–79 days of the dolphins' exposure to a low salinity environment. Dolphins exposed to the lowest salinity levels were more likely to have skin lesions than dolphins exposed to moderate salinity levels. Skin lesions were present in 67% of animals exposed to a salinity level <10 ppt (n = 2 of 3 dolphins); skin lesions were present in 24% of animals exposed to a salinity level between 11 and 20 ppt (n = 4 of 17 dolphins); skin lesions were not observed in animals exposed to a salinity level between 21and 30 ppt (n = 0 of 2 dolphins). Twenty-four dolphins were exposed to a variety of salinity levels encompassing multiple categories. Within this subset, the prevalence of skin lesions was 42% (n = 10 of 24 dolphins; 0–10 ppt, 11–20 ppt, 21–30 ppt).

The most common epidermal change reported in this analysis was a color change to the epidermis (n = 10), followed by abnormal epithelial sloughing (n = 7), and erosive/ulcerative lesions (n = 4) (Figure 2).

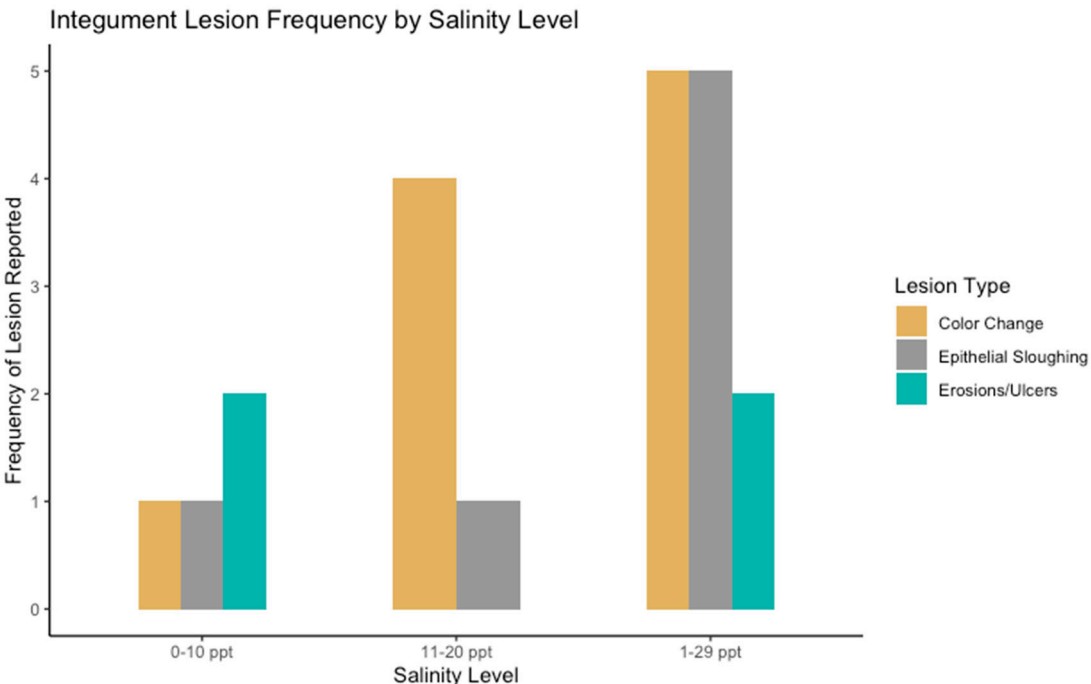

**Figure 2.** The frequency of integument lesions reported by the level of salinity the dolphins were exposed to. Lesions were observed in dolphins exposed to salinity categories of 0–10 ppt (Category 1, sample size n = 3) and 11–20 ppt (Category 2, sample size n = 17). Lesions were also observed in dolphins exposed to a wide range of salinities ranging from 1 to 29 ppt (n = 24). Dolphins that were only exposed to salinities ranging from 21 to 30 ppt (n = 2) did not develop skin lesions.

Of the 16 dolphins who developed skin lesions, mild to moderate lesions (color change and abnormal sloughing) were the only observed lesions in dolphins exposed to a range of salinity levels between 1 and 24 ppt for a total of 13–16 days of consecutive exposure (n = 12). Four dolphins sustained epidermal lesions classified as severe (erosive/ulcerative lesions): two dolphins were exposed to complete freshwater (0 ppt) between 19 and 79 consecutive days, and two dolphins were exposed to a range of salinity levels between 1 and 29 ppt between 77 and 79 consecutive days (average salinity 19 ppt). Mild epidermal lesions formed first, characterized by wrinkling and discoloration of the epidermis, followed by sloughing and further progression into erosions and ulcers of the integument. Of the four dolphins who sustained severe epidermal lesions, two were exposed to freshwater (0 ppt) and the other two were exposed to a wide range of salinity (1–29 ppt)—the first reported epidermal lesion occurred after 17 consecutive days of freshwater exposure (0 ppt) and after 16 consecutive days of exposure to salinities ranging between 1–29 ppt, respectively. Wrinkling of the epithelium around the rostrum and oral commissures has been observed in dolphins exposed to freshwater (0 ppt) as early as 3 days after the first exposure (S. H. Ridgway, pers comm., 1 September 2019). Figure 3 depicts the timeline of epidermal changes from a single dolphin exposed to a wide range of salinities between 1 and 29 ppt for 79 consecutive days.

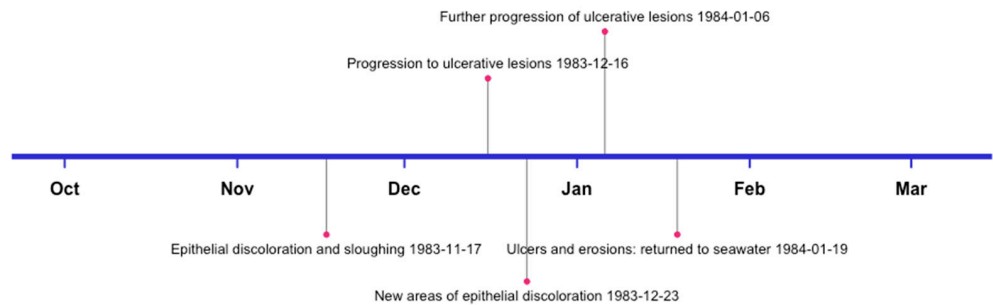

**(A)**

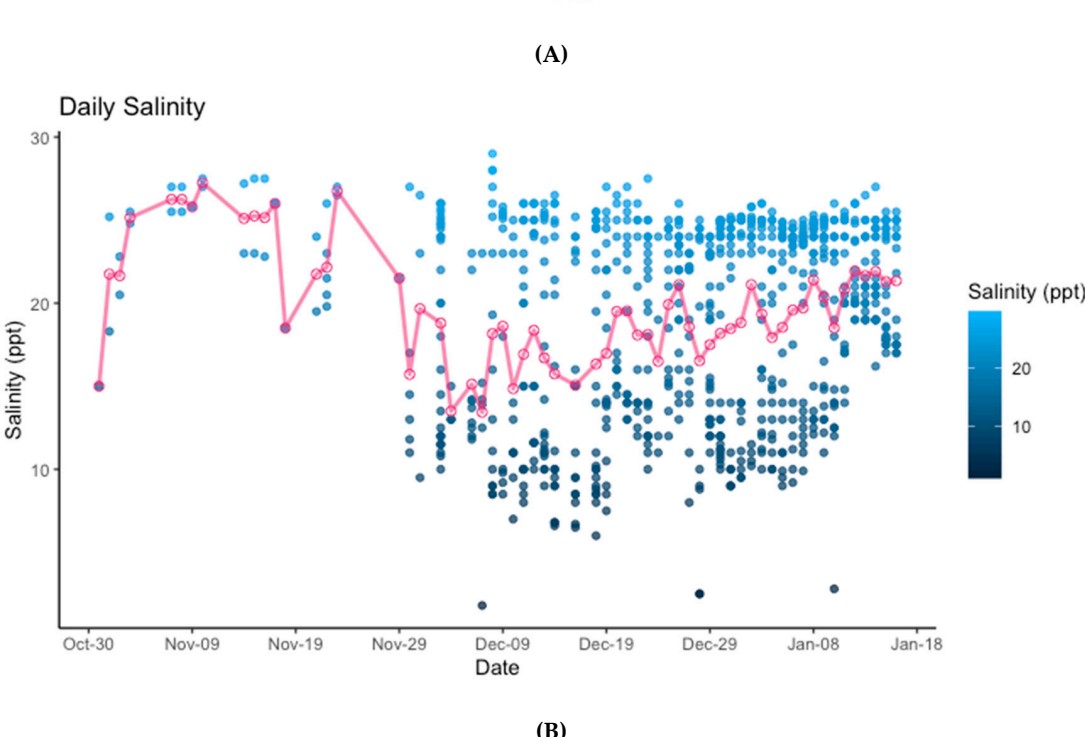

**(B)**

**Figure 3.** (**A**) A timeline of the documented skin lesions observed in an individual dolphin exposed to a salinity range of 1–29 ppt. (**B**) The daily salinity fluctuations are depicted under the timeline. The blue datapoints indicate the actual measured salinity (ppt) at multiple time points each day. The open pink data points and solid line indicate the average daily salinity over the course of time the dolphin spent in low salinity. The total duration of time the animal spent in this environment was 79 consecutive days.

## 4. Discussion

This retrospective analysis found physiologic evidence of systemic changes secondary to low salinity exposure in bottlenose dolphins. On an average, the lower the salinity exposure, the lower the serum sodium, chloride, and osmolality, and the higher the serum BUN. Based on the data available for this study, the level of salinity, rather than the duration of time a dolphin spends in a lower salinity environment, is the main determinant of the magnitude of blood work changes. For the development of skin lesions, both the level of salinity and the duration of exposure to a low salinity environment were important contributing factors. The most serious combination of physiologic effects (severe skin

lesions combined with the lowest serum sodium, chloride, and osmolality and the highest BUN) occurred in dolphins exposed to salinities of <10 ppt.

The retrospective nature of the data and analyses presented multiple potential confounders that were impossible to control. While blood samples were collected via routine methods and processed according to standard procedures, multiple laboratories processed the samples over the course of the 43-year time frame of the study, on many different automated analyzers. As the dolphins involved in this analysis were performing normal, daily tasking and sampling was performed as part of their routine health monitoring and care, samples were collected on both fasting and non-fasting animals and sample collection frequency and timing after the first exposure to a low saline environment were highly variable. The same holds true for the presence of integumental changes secondary to low salinity exposure. Evidence of compromised integrity to the epidermis was based on a written documentation of a change in the appearance of a dolphin's skin. If there was no written documentation, the skin changes could have been missed and in turn it could have reduced the observation of prevalence and timing of skin changes secondary to low salinity exposure. Other uncontrolled and potential confounding variables include water quality and water temperature. The dolphins involved in this study inhabited waters ranging in temperatures from 41.2–80 degrees Fahrenheit and blood samples were collected during every month of the year. Small sample sizes associated with both integument observations and blood samples collected while dolphins were in low salinity waters may have also added additional bias to the results. Therefore, the findings presented here should represent a conservative estimate of the frequency of physiological changes and minimum durations required for them to present.

The health impacts of low salinity exposure on dolphins are likely dependent on multiple variables, including routes of exposure (oral ingestion and/or skin absorption), exposure parameters (duration, salinity profiles), other environmental variables (water quality, water temperature, air temperature), and an individual animal's health status and life history. Studies have shown that dolphins do ingest a small amount of water, either passively through foraging or actively through drinking [9,11]. Changes in serum electrolytes have been observed secondary to ingestion of both seawater and freshwater, and the shifts in serum sodium and chloride observed in this study may suggest water absorption via the gastrointestinal tract, similar to what has been described for marine teleosts [9,11,14]. Serum electrolyte shifts likely stimulate an adrenocortical response, as evidenced by the significant increase in serum aldosterone measured during low salinity exposure. Mammalian aldosterone is utilized in fine-tuning water absorption in the distal tubule and collecting ducts of the nephron by impacting sodium absorption and potassium excretion [34]. Newly published data has shown that urine sodium and chloride output rapidly decrease when dolphins are placed in freshwater (0 ppt), and rapidly increase when dolphins are moved back to a seawater environment [35]. Thus, the increase in serum aldosterone in conjunction with evidence of rapid urine electrolyte output changes, suggesting that serum sodium and chloride are conserved via the kidneys during low salinity exposure in dolphins, similar to what has been documented in baleen whales [9].

Aldosterone has been shown to be secreted as a stress hormone in dolphins, and in response to cold water exposure [13,36,37]. A stress response is considered unlikely in this study as the dolphins were performing routine activities and there was no significant increase in serum cortisol or glucose when measured, both of which have been shown to increase in response to stress in marine and terrestrial mammals [13,37]. An increase in serum aldosterone secondary to cold water exposure is a potential confounder in this analysis as the water temperature was an uncontrolled variable in all events where dolphins were exposed to a lower salinity environment and serum aldosterone samples were collected.

Our findings are consistent with previous observations that healthy dolphins have homeostatic mechanisms to adapt to low salinity conditions, likely including secreting aldosterone from the adrenal gland in order to conserve serum sodium and chloride [3,6–8]. However, dolphins with a compromised adrenal gland may be less likely to adapt to salinity changes in their environment and may be more

susceptible to severe electrolyte derangements. In studies since 2011, dolphins inhabiting Barataria Bay, Louisiana, an estuarine system with highly variable salinity levels, have abnormally low measures of adrenal hormones (cortisol and aldosterone) as well as thin adrenal cortices observed via histopathology on dead stranded dolphins consistent with hypoadrenocorticism [5,38,39]. Barataria Bay was heavily oiled during the Deepwater Horizon oil spill in 2010 and the changes documented in the dolphin adrenal glands are likely secondary to petroleum exposure and toxicity [38]. Due to adrenal compromise, Barataria Bay dolphins as well as other dolphins that have been affected by petroleum, likely have an impaired ability to respond to salinity changes in their environment due to compromised homeostatic mechanisms, ultimately leading to electrolyte derangements and potential detrimental health effects.

Serum sodium and chloride were decreased in dolphins exposed to low saline water compared to controls; however, there were no severe outward clinical symptoms of hyponatremia (e.g., seizures, abnormal mentation) observed in dolphins during these natural exposure events [15]. However, it should be noted that these dolphins were under professional care and their environment was modified if subclinical changes were detected in their blood or skin changes were observed. On an average, healthy dolphins exposed to low salinity waters as low as 0ppt can likely regulate serum electrolyte homeostasis via an adrenocortical response and reabsorption of electrolytes via the renal system for multiple days, even weeks with only mild serum electrolyte disturbances observed. In a recently published study, a single free-ranging dolphin was monitored and rescued from a freshwater lake after 32 days of exposure and had mild hyponatremia (146 mEq/L) and hypochloremia (98 mEq/L) but there was no evidence of clinical symptoms secondary to the electrolyte derangements [3]. From the data in the current analysis, two dolphins spent 19 and 79 consecutive days in freshwater (0 ppt), one of which had multiple blood samples drawn during freshwater exposure. Fifty-nine days after the first exposure to freshwater, the last blood date on record, a mild hyponatremia (149 mEq/L) and hypochloremia (100 mEq/L) (normal reference range of 153–159 mEq/L and 116–124 mEq/L, respectively) [40] were observed, with no outward systemic illness secondary to these electrolyte derangements. However, dolphins with underlying adrenal or renal disease will likely have a compromised response to the changes in serum electrolytes. Gastrointestinal symptoms of nausea, vomiting, and diarrhea are characteristic of mild hyponatremia in humans [15]. A recently published report [35] showed that a bottlenose dolphin with severe renal compromise had systemic clinical signs of diarrhea and vomiting secondary to a rapid shift in serum sodium within twelve hours of exposure to freshwater. The dolphin was placed in freshwater as part of a medical treatment plan for severe renal compromise. Serum sodium levels decreased from 154 mEq/L (measured immediately prior to freshwater exposure and within the typical reference range of 153–159 mEq/L [40]) to 149 mEq/L (measured after twelve hours of freshwater exposure). Upon returning to a marine environment (35 ppt), the measured serum sodium was within the normal reference range (155 mEq/L) and the diarrhea and vomiting had resolved within 24 hours [35].

An interesting finding in this analysis was the significant elevation in serum BUN, but not creatinine, in dolphins exposed to lower salinity (compared to reference dolphins). Further research is required to understand the pathophysiology and clinical significance of this finding. Elevated serum BUN can be secondary to renal and non-renal mechanisms [41]. With the presence of normal serum creatinine, a renal etiology is considered unlikely in this study. The primary non-renal etiologies causing an increased serum BUN are aging and an increased protein level (e.g., increased protein consumption, increased muscle breakdown, and gastrointestinal hemorrhage) [41,42]. In our analysis, age did not have an effect on BUN. An increased protein level secondary to increased protein consumption and/or increased muscle breakdown is considered unlikely as the diet and weight of the dolphins involved in this analysis were unchanged during exposure to low salinity environments. Mild gastrointestinal hemorrhage cannot be ruled out in this study as gastrointestinal diagnostics like endoscopy and stomach sampling were not performed during these deployments; however, evidence of severe gastrointestinal hemorrhage like melena was not present. In other species like elasmobranchs, urea is conserved to maintain plasma osmolality, a phenomenon also observed in humans with renal failure

to counteract the fall in plasma osmolality secondary to hyponatremia [9,15]. It is possible that the increase in serum BUN observed in this study was a counteractive measure to maintain the serum osmolality in the face of a decreased serum sodium level.

The majority of dolphins that spent time in <10 ppt salinity environments developed skin lesions (67%). Additionally, 42% of dolphins exposed to a wide range of salinity levels encompassing multiple salinity categories developed skin lesions. Dolphins that spent longer periods of time in low salinity water developed more severe skin lesions (ulcerative/erosive), indicating that both the salinity level and the duration of time a dolphin spends in a low salinity environment play a role in the development and severity of skin lesions. Resolution of severe skin lesions observed in one dolphin that was exposed to salinities ranging from 1–29 ppt for 79 consecutive days occurred over the course of a 90-day period after returning to a marine environment. Deming et al. [3] found that ulcerative skin lesions had almost completely resolved 85 days after a dolphin was relocated to an environment with higher salinity (average of 13.7 practical salinity units). As this was not a controlled study, early signs of skin lesions may not have been documented immediately. Other uncontrolled variables that may have contributed to the development of skin lesions in this analysis include water and air temperature, water quality, and potential sun exposure [21].

Water absorption through the skin has been considered unlikely under normal circumstances in bottlenose dolphins; however, if the protective barrier of the skin has been compromised, water absorption would likely occur leading to a net increase in total body water [9,11]. The effects of compromised integument on the electrolyte changes in this study were unable to be characterized due to the variable timing of blood sample collection and epidermal assessment and documentation. In some dolphins, electrolyte changes were observed within 24 h of low salinity exposure with no evidence of skin lesions and in other dolphins, mild to moderate skin lesions were observed after 13 days of low salinity exposure with no evidence of electrolyte changes during this time. These findings neither prove nor disprove skin absorption as a potential route of freshwater exposure and electrolyte imbalance and provide further evidence that the physiological effects of low salinity on bottlenose dolphins are likely multifactorial.

Future work is necessary to fully understand the effects and thresholds of low salinity exposure on bottlenose dolphins. Collaborative efforts between stranding networks and managed care facilities should include as many diagnostics and environmental variables when responding to dolphins in low salinity environments. Data points could include the water salinity at the stranding site and the duration of time the dolphin(s) were exposed to the specific salinity level; serial blood and urine monitoring for specific gravity, electrolytes and osmolality; serum hormone analysis; morphometrics and a full physical examination of the dolphin; photographs with measurements, skin scrapings, and biopsies of any skin lesions present; and the placement of a satellite tag for tracking upon release. Data points for deceased dolphins could also include cerebrospinal fluid, vitreous, and aqueous humor (NOAA Freshwater Exposure Sampling Plan). The physiological changes observed in this analysis improve our understanding of the upper limit of duration and the lower limit of salinity in which a dolphin can maintain homeostasis in a low salinity environment. With the increasing threat of exposure to abnormally low saline waters on free-ranging dolphins secondary to climate change, increased rainfalls, and other anthropogenic factors, there is an urgent need for a more comprehensive understanding of the physiologic effects of low salinity on bottlenose dolphins.

**Author Contributions:** Each author made substantial contributions to the creation of this manuscript. Conceptualization, S.H.R. and C.R.S.; methodology, A.M.M. and R.T.; validation, R.T.; formal analysis, A.M.M. and R.T.; investigation, S.H.R., R.D. and A.M.M.; resources, E.D.J., S.H.R. and R.D.; data curation, R.D., A.M.M. and S.H.R.; writing–original draft preparation, A.M.M.; writing–review and editing, R.T., S.H.R., R.D., F.M.G., C.R.S. and E.D.J.; supervision, E.D.J.; visualization, A.M.M.; funding acquisition F.M.G. and C.R.S. All authors have read and agree to the published version of the manuscript.

**Funding:** This research was funded by The Department of Commerce: NOAA, subcontracted through the Water Institute of the Gulf under federal award number NA19NMF4630074.

**Acknowledgments:** The authors would like to sincerely thank Ashley Barratclough for her significant time and effort that was put into making this publication a reality. The authors also sincerely thank the U.S. Navy Marine Mammal Program for providing the highest standard of care to the Navy dolphins over the past 60 years and collecting such valuable data. This is scientific contribution number 282 of the National Marine Mammal Foundation.

**Conflicts of Interest:** The authors declare no conflict of interest. The funders had no role in the design of the study; in the collection, analyses, or interpretation of data; in the writing of the manuscript, or in the decision to publish the results.

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
