# Peer review of "Physiological Effects of Low Salinity Exposure on Bottlenose Dolphins (Tursiops truncatus)"

_2673-5636, doi:10.3390/jzbg1010005_

Round 1

Reviewer 1 Report

This research describes a novel approach for assessing the effects of low salinity environments on bottlenose dolphins by testing blood biochemistry and visually evaluating skin condition. This work is important and timely, particularly considering climate-induced changes in a variety of dolphin habitats. Overall, the work is well-described; however, as with any longitudinal retroactive dataset, there are a number of factors that could not be controlled or standardized. Many of these shortcomings are discussed by the authors, and I have made suggestions/inquiries about other potential covariates (embedded as comments in the attached text). I also suggest considering a multivariate statistical approach, particularly for testing blood biochemistry.

Please see the attached document for specific feedback. Very broadly, I have suggested the following:

Several examples of superfluous wording can be condensed.

Methods

Salinity categories should not contain the same number twice. For example, 0-10ppt, 10-20ppt… Consider revising to something like 0-10ppt, 11-20ppt…

Why are covariates such as age/age class/sexual maturity/sex/hormonal cycle/etc, not included as potential explanatory factors? Particularly in a study involving known animals. If demographic factors such as these are not known to affect blood values, consider including in the text. 

While specific geographic locations may not be necessary, it would be helpful to include basic location and water temperature information, if possible. For example, were samples from estuaries, bays, highly urbanized areas, remote locations, etc. Were they collected in cold or warm temperature waters? More information would be helpful to consider potential explanatory factors. If these factors are not known to affect blood values, consider including in the text.

Although data were collected retroactively, can time of year or water temperature categories be included

I suggest considering a multivariate statistical approach for each blood test that can incorporate multiple potential covariates and interactions.

Results

Several acronyms in the tables should be spelled out so that they are stand-alone from the main text.

I suggest including the separate significant salinity categories in Figure 1, rather than lumping all low salinity values together.

Figure 3 diagrams could be standardized for better visual comparison – it is difficult to see how they relate at first glance.

Discussion

Low sample sizes and associated limitations should be discussed.

A broad discussion on how findings from the blood analyses and the visual skin condition may relate to one another (or are unable to be related at this stage, and suggestions for future work) considering these are the two overarching factors evaluated. Can these findings complement each other?

Literature

I suggest taking a look at the following references on anomalous skin conditions:

Hart, L. B., Rotstein, D. S., Wells, R. S., Allen, J., Barleycorn, A., Balmer, B. C., ... & McFee, W. (2012). Skin lesions on common bottlenose dolphins (Tursiops truncatus) from three sites in the Northwest Atlantic, USA. PLoS One7(3), e33081.

Mullin, K., Barry, K. P., Sinclair, C., Litz, J. A., Maze-Foley, K., Fougères, E. M., ... & Tumlin, M. (2015). Common bottlenose dolphins (Tursiops truncatus) in Lake Pontchartrain, Louisiana, 2007 to mid-2014.

Toms, C. N., Stone, T., & Och‐Adams, T. (2020). Visual‐only assessments of skin lesions on free‐ranging common bottlenose dolphins (Tursiops truncatus): Reliability and utility of quantitative tools. Marine Mammal Science.

Reviewer 2 Report

REVIEW

General comments:

This article assesses the physiological effects of low salinity exposure on bottlenose dolphins (Tursiops truncatus). The article is generally well-written and worth-publishing. The data analysis methodology is well chosen. I would suggest re-writing a paragraph in the result section (epidermal assessment) as some parts were unclear to the reader. I also have some comments and suggestions below.

Line 134-136: Could the authors explain why the upper category was chosen? 

Line 139-141: "Visual epidermal assessment was performed on all dolphins: inclusion criteria for assignment to a specific salinity category required the dolphin to have spent the entirety of the low salinity exposure period in one of the three salinity categories (0-10 ppt, 10-20 ppt, 20-30 ppt)."

Could the author re-write this sentence? The reader is unsure to fully understand the meaning. What does the author mean by "entirety of the ... exposure period"?

Line 171-172

“Blood values obtained while dolphins were in lower salinity environments were 172 compared to values obtained while dolphins were in marine environments (controls)”

Could the author specify the salinity range dolphins were exposed to in marine environments?

Line 189 (legend of figure 1): “normal marine environment”. I would suggest using the word “control”, as mentioned by the authors line. 172, instead of “normal”, as “normal” can vary depending on locations/periods... (same comment apply to line 193, 198, 202, 206, 210, 214, 225…).

Line 195… there is a symbol before “= 15.91”. Could you specify the test used and what this symbol is? (same comment lines 204/207/209/211)

Line 239: “lower salinity environment”. Lower than?

Paragraph line 238 (Epidermal assessment):

As commented above, the reader does not understand the period of time the dolphins were exposed to a particular salinity exposure.

Could the author re-write the end of the sentence (239-240) specifying the time frame?

: “… for any amount of time.”

The reader does not understand if the presence of skin lesions were assessed before the exposure to low salinity environment or only after?

Could the author precise the total number of dolphins for which the epidermal assessment was done? In the first sentence of the paragraph (line 239), the author cites skin changes in 16 dolphins exposed to lower salinity. Then, the author mentioned skin lesions in 3 dolphins in salinity <10ppt and 17 dolphins in salinity 10-20ppt. The total is now 20 dolphins. Are the 16 dolphins cited above part of these 20 dolphins? Could the author specify?

Line 239: the author mentioned skin “changes”. Does that mean that skin lesions assessments were done before and after the exposure to a lower salinity level for 16 dolphins? Could the author specify the time frame?

From line 241-244: the author only mentioned the presence of skin lesions (not mentioning anymore the word “changes”) for 20 dolphins. Could the author specify?

Lines 247-249: the author mention “epidermal change”. Does this paragraph apply only to the 20 dolphins mentioned lines 241-244? Or to all the dolphins? Please specify.

Figure 2: in the legend, the author mention “Lesions were also observed in dolphins exposed to a wide range of salinities 254 ranging from 1-24 ppt”, but on the figure, the salinity range (right side) is 1-29 ppt?

Could the author cite the sample size (number of dolphins) for each of the categories?

Lines 263-266: “The first reported epidermal change in the dolphins exposed to freshwater (0 ppt) occurred 17 days after the first exposure to freshwater and the first reported epidermal change in the dolphins exposed to a salinity range between 1-29 ppt occurred 16 days after the first exposure to low salinity.”

The reader does not understand if the dolphins in these cases were exposed during the full 17 and 16 days to freshwater/low salinity or if they were exposed for a shorter period of time and the skin lesions appeared 17 and 16 days after, respectively. Please specify.
